# Dynamic, Transient, and Robust Increase in the Innervation of the Inflamed Mucosa in Inflammatory Bowel Diseases

**DOI:** 10.3390/cells10092253

**Published:** 2021-08-30

**Authors:** Miguel Gonzalez Acera, Marvin Bubeck, Fabrizio Mascia, Leonard Diemand, Gregor Sturm, Anja A. Kühl, Raja Atreya, Dieter Chichung Lie, Markus F. Neurath, Michael Schumann, Christoph S.N. Klose, Zlatko Trajanoski, Christoph Becker, Jay V. Patankar

**Affiliations:** 1Department of Medicine 1, University of Erlangen-Nuremberg, 91052 Erlangen, Germany; Miguel.GonzalezAcera@uk-erlangen.de (M.G.A.); Marvin.Bubeck@uk-erlangen.de (M.B.); Fabrizio.Mascia@uk-erlangen.de (F.M.); leonarddiemand@gmail.com (L.D.); raja.atreya@uk-erlangen.de (R.A.); Markus.Neurath@uk-erlangen.de (M.F.N.); christoph.becker@uk-erlangen.de (C.B.); 2Deutsches Zentrum Immuntherapie (DZI), 91054 Erlangen, Germany; 3Biocenter, Institute of Bioinformatics, Medical University Innsbruck, 6020 Innsbruck, Austria; mail@gregor-sturm.de (G.S.); zlatko.trajanoski@i-med.ac.at (Z.T.); 4iPATH.Berlin Histopathology Core Unit, Charité-Universitätsmedizin Berlin, Corporate Member of Freie Universität Berlin, Humboldt Universität zu Berlin, and Berlin Institute of Health, 12203 Berlin, Germany; anja.kuehl@charite.de (A.A.K.); 5Department of Gastroenterology, Infectious Diseases and Rheumatology, Charité-Universitätsmedizin Berlin, Corporate Member of Freie Universität Berlin, Hindenburgdamm 30, 12200 Berlin, Germany; michael.schumann@charite.de; 6Institut für Biochemie, Emil-Fischer-Zentrum, Friedrich-Alexander-Universität Erlangen-Nürnberg, 91054 Erlangen, Germany; chi.lie@fau.de; 7Department of Microbiology, Infectious Diseases and Immunology, Charité-Universitätsmedizin Berlin, Corporate Member of Freie Universität Berlin, Humboldt-Universität zu Berlin, Hindenburgdamm 30, 12203 Berlin, Germany; christoph.klose@charite.de

**Keywords:** enteric nervous system, neurogenesis, inflammatory bowel diseases, ulcerative colitis, Crohn’s disease

## Abstract

Inflammatory bowel diseases (IBD) are characterized by chronic dysregulation of immune homeostasis, epithelial demise, immune cell activation, and microbial translocation. Each of these processes leads to proinflammatory changes via the release of cytokines, damage-associated molecular patterns (DAMPs), and pathogen-associated molecular patterns (PAMPs), respectively. The impact of these noxious agents on the survival and function of the enteric nervous system (ENS) is poorly understood. Here, we show that in contrast to an expected decrease, experimental as well as clinical colitis causes an increase in the transcript levels of enteric neuronal and glial genes. Immunostaining revealed an elevated neuronal innervation of the inflamed regions of the gut mucosa. The increase was seen in models with overt damage to epithelial cells and models of T cell-induced colitis. Transcriptomic data from treatment naïve pediatric IBD patients also confirmed the increase in the neuroglial genes and were replicated on an independent adult IBD dataset. This induction in the neuroglial genes was transient as levels returned to normal upon the induction of remission in both mouse models as well as colitis patients. Our data highlight the dynamic and robust nature of the enteric nervous system in colitis and open novel questions on its regulation.

## 1. Introduction

The concerted activity of multiple cell types in the gut ensures that homeostasis is maintained. In recent years, crucial functions of the enteric nervous system (ENS) have been uncovered that help in maintaining this homeostasis. Extrinsic and intrinsic sensory innervations of the gut mucosa and enteric glia can sense and respond to changes in the intestinal chemo-microbial milieu [1,2,3,4]. Therefore, alterations in the gut microbial-immune dynamics, which underlies certain chronic intestinal inflammatory conditions such as inflammatory bowel diseases (IBD), are expected to affect the function of the ENS. Key experiments from germ-free mice have revealed that commensal queues are critical in the differentiation of a normal ENS [5]. Interestingly, these regulatory influences are not restricted to the developing gut as antibiotic-mediated depletion of gut bacteria in adult life recapitulates the poor neuronal differentiation seen in germ-free animals. The microbial diversity is greatly impacted during colitis, and antibiotic therapy is widely used as a first-line therapeutic strategy in IBD. However, little is known about how these processes impact the ENS.

The ENS is being increasingly recognized for its role in modulating intestinal immune responses [6,7]. A cryptic leakiness in the intestinal barrier and/or auto-activation of the immune cells due to causes that are still elusive is envisaged to underlie the triggering mechanism in IBD. Invading microbes and proinflammatory cytokines lead to further breakdown of the epithelial barrier via the induction of intestinal epithelial cell (IEC) death release of damage-associated molecular patterns (DAMPs) into the lamina propria [8,9]. Recent breakthrough research using single-cell transcriptomic approaches has revealed the diversity in the expression of key IBD-related cytokine and pattern recognition receptors on the neurons and glial cells of the ENS in healthy conditions [10,11,12,13]. However, it is unclear how the expression of these receptors changes the behavior of the cells of the ENS during pathological and inflammatory states such as colitis. Furthermore, it is unclear whether the ENS survives the proinflammatory insults or is lost due to cell death like other parenchymal cells in the gut tissue such as the IECs [4,14,15,16]. Some previous studies have addressed whether cell death occurs in the ENS in homeostasis and disease with inconclusive results. Kulkarni et al. showed that there is an active turnover of enteric neurons in the gut during homeostasis, whereas Joseph et al. using various mouse models showed that detection of BrdU-positive neurons was inconclusive in assessing neurogenesis during intestinal inflammation [17,18]. Taken together, these data highlight the need for further research into the fate and function of the ENS, which will improve our understanding of inflammatory bowel disease biology.

Here, using various mouse models, each representing a wide sub-pathology of IBD, we investigate the changes that occur in the ENS during colitis. We use transcriptomic, molecular biological, and immunohistochemical approaches and identify that in colitis-associated with overt epithelial damage, the ENS is amplified with greater projection into the inflamed tissue. Conversely, deep ulceration of the tissue was associated with an overall depletion of the underlying myenteric plexi and a local loss of the ENS. Taken together, our data highlight the dynamic nature of how the ENS responds to different inflammatory insults in colitis and provides a framework on the suitability of various mouse models to investigate the ENS in colitis.

## 2. Materials and Methods

### 2.1. Mouse Housing and Husbandry

All mice were housed in individually ventilated cages, in specific-pathogen-free facilities under equal light-dark conditions and had ad libitum access to standard mouse chow diet and water. The experimental mice were randomly assigned to test or control groups and consisted of both sexes. The mice were between the ages of 8 to 15 weeks during the experiments. All experiments with mice were performed upon prior approval based on the ethical regulations of the institutional review board and the ethics committee of the University of Erlangen-Nürnberg and the ethics commission of Lower Franconia (55.2 2532–2-595, -607, -1137, -1178).

### 2.2. Experimental Models of Murine Colitis

All experiments described in this study were conducted on C57BL6 or Rag1^-/-^ mice between 8 and 18 weeks of age, which were purchased from Charles River Laboratories (Sulzfeld, Germany). The protocols for the establishment of various mouse colitis models and the transcriptomic analysis of the colon tissues for the various mouse models have been previously described [19,20,21,22]. Briefly, acute colitis was induced by challenging mice with 3% dextran sulfate sodium (DSS) in drinking water for 8 days, whereas chronic was induced by two 1-week rounds of 1.5% DSS in drinking water interspersed with a 2-week break on normal sterile drinking water. For the induction of oxazolone-induced colitis, the mice were anesthetized and pre-sensitized by the application of 150 µL of a 3% solution of oxazolone in a 4:1 acetic acid to oil mixture on the shaved upper back skin. On day 8 post pre-sensitization, the mice were anesthetized and received a rectal enema of 100 µL of a 0.5% oxazolone solution. The mice were kept in a head-down position for 1 min after administering the enema. Adoptive T cell transfer colitis was initiated in Rag1^-/-^ mice by intraperitoneally injecting 5 × 10^5^ of CD4^+^, CD25^−^, and CD45RB^high^ splenic naïve effector T cells sorted from wild-type C57BL6 mice.

### 2.3. RNA Extraction

The respective colon tissue samples from various mouse models were subjected to total RNA isolation using the microspin total RNA kit (VWR International GmbH, Langenfeld, Germany) as per the manufacturer’s protocol. The quality of the isolated RNA was controlled for degradation and contamination using 1% agarose gels. Further quantification and quality controls were performed using the NanoPhotometer^®^ spectrophotometer (Implen GmbH, Munich, Germany), Nanodrop (Thermo Fisher Scientific GmbH, Erlangen, Germany), Qbit (Thermo Fisher Scientific GmbH, Erlangen, Germany), and the RNA Nano 6000 Assay Kit of the Bioanalyzer 2100 system (Agilent Technologies Deutschland GmbH, Waldbronn, Germany).

### 2.4. Library Preparation and mRNA Sequencing

A total amount of 1 μg RNA per sample was used as input material and sequencing libraries were generated using NEBNext^®^ UltraTM RNA Library Prep Kit for Illumina^®^ (New England Biolabs, Ipswich, MA, USA) following the manufacturer’s recommendations. Molecular index codes were added at this step to allow sample tagging. The mRNAs were purified from total RNAs using poly-T oligo-attached magnetic beads followed by fragmentation via divalent cations and elevated temperature in NEBNext First Strand Synthesis Reaction Buffer (5X). First-strand cDNA was synthesized using random hexamer priming and M-MuLV Reverse Transcriptase (RNase H-). Second-strand cDNA synthesis was performed using DNA polymerase I and RNase H. The AMPure XP system (Beckman Coulter, Brea, CA, USA) was used to select cDNA fragments of 150–200 bp. To assist with directional RNA sequencing, uracil-specific excision reaction was performed using 3 μL USER Enzyme (NEB, USA) at 37 °C for 15 min followed by 95 °C for 5 min. PCR was performed using the Phusion High-Fidelity DNA polymerase, Universal PCR primers, and Index primers. At last, PCR products were purified using the AMPure XP system(Beckman Coulter, Brea, CA, USA) and library quality was assessed on the Agilent Bioanalyzer 2100 system.

The index-coded samples were clustered on a cBot Cluster Generator using PE Cluster Kit cBot-HS (Illumina, San Diego, CA, USA) according to the manufacturer’s instructions. After cluster generation, paired-end sequence reads were generated by sequencing the library on an Illumina platform. Clean reads were obtained and GC content of the clean data was calculated after adapter trimming and removal of reads with low quality and with poly-N sequences. Only the clean reads generated from this step with high quality were used in all the downstream analyses. Reference genomes and gene model annotation files were downloaded from NCBI, UCSC, and Ensembl directly. Paired-end clean reads were mapped to the reference mouse genome (GRCm38.p6) using the STAR software [23] (2.7.0d).

### 2.5. Single-Cell RNA-Seq-Assisted Generation of the Neuroglial Identifier Lists

Neuroglial identifier gene lists were generated taking advantage of the newly available single-cell RNA sequencing datasets from the enteric nervous system (SCP1038 [10], SRP135960 [12]), intestinal epithelial and stromal cells (GSE92332 [24], SCP259 [25]), as well as bulk RNA sequencing datasets from isolated intestinal epithelial organoids (GSE148727 [26], GSE159423 [27]). Genes with a specific expression in the enteric neuroglial datasets but not the other datasets including epithelial and stromal were chosen to create the list. The list was cross-referenced with available literature for expression in neurons and glial cells [28,29,30,31].

### 2.6. Analyses of Publicly Available Datasets

We used previously published transcriptomic datasets from PROTECT [32] (accession: GSE109142, 206 samples from ulcerative colitis, and 20 samples from non-inflamed rectum tissues) and Fenton et al. [33] (accession: GSE128682, 16 control samples from normal colon tissues, 14 samples from active ulcerative colitis colon tissues, and 14 samples from colon tissues from ulcerative colitis patients in remission) cohorts. The gene expression changes in the neuroglial identifiers established in our study were evaluated.

### 2.7. Statistical Analysis of Differential Gene Expression

We used FeatureCounts (v2.0.1) to call the read numbers for each gene mapped to the Ensembl database. Differential expression between two groups was performed using the DESeq2 (v 1.26.0) R package [34]. DESeq2 uses a model based on the negative binomial distributions and provides statistical routines for determining differential expression in RNA-Seq data [34]. The resulting *p*-values were adjusted using Benjamini and Hochberg’s approach for controlling the false discovery rate (FDR). Genes with adjusted *p* < 0.05 found by DESeq2 were assigned as differentially expressed. Gene sets for neurons and glia were defined based on the neuronal and glial identifiers as above. All significances were called based on adjusted *p*-values < 0.05.

### 2.8. Immunohistological Staining and Analyses

Tissues were fixed in 4% paraformaldehyde and embedded in paraffin. Tissue sections of 5–10 µm in thickness were subjected to deparaffinization using a solvent series. The tissue sections were subjected to heat-induced antigen retrieval in a Tris-EDTA buffer (pH = 9.0). The tissues were then washed, blocked, and incubated overnight with the biotinylated primary antibody directed against TUBB3 (1:100, Biolegend, Cat#: 801212, clone: TUJ1) and the Alexa-488-conjugated antibody against GFAP (1:100, Biolegend, Cat#: 644744, clone: 2E1.E9). The sections were washed with 1X PBST thrice and subjected to TUBB3 staining, followed by an incubation step in Streptavidin-Dylight555 conjugates (1:500, DyLight, Invitrogen) for 1 h. Excess unbound streptavidin was washed using 1X PBS and nuclei were counterstained using Hoechst 33,342 (Thermo Fisher Scientific GmbH, Erlangen, Germany). The samples were mounted using a fluorescence mounting medium (Agilent Technologies Deutschland GmbH, Waldbronn, Germany) and imaged on the Leica SP5 confocal microscope. Quantification of TUBB3- and GFAP-positive areas was performed on single-channel threshold grayscale images in Fiji (Image J2) and expressed as a percentage of the total area.

## 3. Results

### 3.1. Experimental Colitis Causes an Increased Neuroglial Repertoire

To address the impact of gut inflammation on the enteric neuroglia, we first generated a list of neuroglial identifier genes that are known to be expressed in the enteric nervous system. To do this, we took advantage of the recently published single-cell RNA sequencing data from the mouse enteric nervous system and the mouse intestine as a whole and chose identifiers that were detectable in the ENS datasets but not in epithelial, stromal, or immune datasets. Next, we analyzed the behavior of these neuroglial identifiers on transcriptomic data in an mRNA sequencing experiment representing different stages of experimental colitis in mice. The different colitis stages represented colon tissues from healthy, mildly inflamed, highly inflamed, moderately resolved, and fully resolved as recently reported [19,20]. The analysis showed that several genes for neurons and glia were misregulated during the various stages of colitis (Figure 1A,B). Notably, we saw a significant induction in the neuroglial identifiers at the peak of DSS-induced inflammation (highly inflamed group; Figure 1A,B). For several genes, we observed a clear, gradual upregulation from healthy to mildly inflamed to highly inflamed (Figure 1A,B). This was more directly visualized by plotting the identifiers in a volcano plot, revealing that the proportion of neuronal genes that were upregulated in the mildly inflamed group was 40%, which doubled to 80% in the highly inflamed group. A similar trend was observed for glial cell identifiers where the proportion of upregulation went from 30% (mildly inflamed) to 54% (highly inflamed). Interestingly, there was a strong reduction in the expression of these genes during the resolution phases with a significant downregulation already in the moderately resolved group indicating the dynamic nature of this response (Figure 1A,B).

To our surprise, the analysis of Reactome pathways from the transcriptome of the highly inflamed versus healthy colon tissues revealed a significant enrichment of pathways for axon guidance and neuronal systems (Figure 1C). Further gene ontology analysis for cellular components, which are upregulated in the highly inflamed colon tissues, revealed an increase in several neuronal processes such as pre- and postsynapse, synaptic and postsynaptic membrane, and multiple other neuronal projections and axonal processes (Figure 1D). These data indicate that intestinal inflammation shares a large component of transcriptomic changes to the enteric nervous system with those seen in neurodegenerative disorders.

### 3.2. Increase in the Neuroglial Repertoire Is Accompanied by Heightened Innervation of the Inflamed Mucosa and Higher Barrier Disruption

To visualize whether the changes detected in the transcriptomic data also corresponded with actual changes in the number, density, and distribution of enteric neuroglia, we immunostained neurons and glia from the colons of mice with or without DSS-induced colitis. The analysis of the immunostaining revealed that there were a greater number of densely packed myenteric plexi in the inflamed tissues compared with controls (Figure 2A). As a consequence of this increased density of myenteric plexi, the interplexial distance was significantly reduced in the inflamed versus the healthy colonic tissues (Figure 2B). Besides the closer interplexial distance, the abundance of neuronal tracks was clearly increased in the inflamed regions of the mucosa innervating the inflamed regions. Furthermore, a significant increase in the areas that were positive for the neuronal protein TUBB3 (+2.9-fold) and the glial fibrillary acidic protein GFAP (+2.3-fold) was detectable in the inflamed tissues (Figure 2D,E). These data indicate that the density and distribution of the enteric neuronal and glial cells located at the myenteric plexi are increased at inflamed regions of the gut and that either intrinsic and/or extrinsic innervation of the inflamed mucosal foci is elevated in experimental colitis models.

To address whether generalized inflammation of the colon suffices to cause the induction in the neuroglial repertoire or whether the mode of colitis induction influences this process, we analyzed transcriptomes from various mouse models of colitis. These included an independent set of acute DSS-induced colitis, oxazolone-induced colitis, and T cell adoptive transfer colitis. Interestingly, compared with acute DSS-induced colitis, all the other models tested showed much milder effects on the changes in the expression levels of the selected enteric neuroglial identifiers (Figure 2F,G). The magnitude of induction in the enteric neuroglial identifiers followed an overall trend, which was, acute DSS-induced colitis > adoptive T cell transfer colitis > oxazolone-induced colitis (Figure 1F,G). These data indicate that the enteric neuroglial expansion occurs in experimental colitis models involving acute barrier damage or T cell-mediated autoimmunity.

### 3.3. The Neuroglial Repertoire Is Increased in the Inflamed Tissues of IBD Patients

Few previous studies have reported IBD-associated enteric neuropathies and, given the strong increase in the neuroglial repertoire detected in experimental colitis models, we recognized that this effect might be underappreciated in human IBD settings. To test whether a similar phenomenon may occur in IBD patients, we investigated IBD transcriptomic datasets. For this, we analyzed two publicly available IBD transcriptomic datasets: (a) the treatment of a naïve pediatric ulcerative colitis cohort (PROTECT [32]), representing patients from both sexes with active UC as well as control patients without active ulcerative colitis; and (b) a dataset representing adult active UC and UC in remission and controls (Fenton et al. cohort) [33]. Interestingly, consistent with our observations on the experimental colitis, UC patients also showed an increased expression in the genes for neuroglia (Figure 3A). This induction in the expression levels of multiple neuroglial identifiers was independent of sex and was highly correlated with inflammation, assessed as Pearson’s correlation with the expression of S100A8, a well-established marker of intestinal inflammation (Figure 3A). The expression of most of the identifier genes was also negatively correlated with that of VIL1, which codes for the epithelial protein villin, indicating that epithelial loss is negatively correlated with increased expression of neuroglial identifiers (Figure 3A). An overall upregulation was detectable in 38% of the identifiers tested, while only 9% were downregulated (Figure 3B). The expression of only three identifier genes in the PROTECT cohort SYP, MBP, and GMFB was negatively correlated with the expression of S100A8 (Figure 3A).

Analysis of a second publicly available dataset showed a similar induction in the expression levels of neuroglial identifiers in the active UC group (Figure 3C), essentially replicating our findings from the PROTECT cohort and reconsolidating the findings from the experimental colitis. Interestingly, the expression pattern of the neuroglial identifiers in the remission group closely correlated with controls, indicating that once inflammation subsides, the expression levels of the neuroglial identifiers in the gut return to baseline levels. This was identical to our findings from the resolved groups of the DSS-induced colitis time course experiments (Figure 1A,B). Surprisingly, the same three identifier genes SYP, MBP, and GMFB, which were negatively correlated with the expression of S100A8 in the PROTECT cohort, were also negatively correlated with S100A8 in the UC remission cohort. Finally, staining inflamed and non-inflamed tissues from UC and Crohn’s disease (CD) patients revealed an increased positivity for TUBB3 in inflamed mucosal tissues compared with non-inflamed tissues (Figure 3D,E). We then analyzed the expression levels of the neuroglial identifiers on an RNA Seq experiment from our own cohort of ulcerative colitis and control patient samples. The analysis showed an upregulation of 36% of the identifiers in the ulcerative colitis groups compared with controls (Figure 3F). Taken together, our data highlight the dynamic nature of the ENS during intestinal inflammation and its resolution and reveal the under-recognized role of the ENS during inflammatory bouts of IBD.

## 4. Discussion

IBD patients frequently experience a reduction in the neuronally regulated functions of the gut such as reduced peristalsis, abdominal pain and cramping, and diarrhea. The inflammatory changes in the gut of IBD patients and the consequent oxidative stress have long been thought to trigger damage to the enteric neuronal tissues manifesting as adverse symptoms experienced by the patients. Previous studies on mouse models have been largely inconclusive on the behavior of the ENS during homeostasis and inflammation of the gut. Despite several contradictory reports, very few studies have actually compared neuroglial composition and ENS architecture in the inflamed gut of mouse models of colitis and from inflamed tissues of inflammatory bowel disease patients. The recent discovery of the ability of enteric neurons and glia to regulate immune functions in the gut has triggered a new wave of investigations into the nature of the triaxial communication between the epithelial barrier-immune and neuroglia compartments in the context of IBD. These include the key discovery of the neuropeptide neuromedin U to act on type 2 innate lymphoid cells and regulate their function during nematode infection [6]. However, the implication of these findings for IBD remains to be investigated. Interestingly, our analysis showed that colitis is associated with a generalized increase in the populations of a variety of neuronal subpopulations. We observed increased expressions of (i) *Nos1*, which is a marker for inhibitory motor neurons, (ii) *Slc5a7* and *Slc17a6*, which are markers for cholinergic neurons, and (iii) *Nmu*, which is a marker of a subpopulation of enteric sensory neurons [10,12,13]. Since our data reflect tissue-level transcriptomes, we found interesting changes in the expression level of *Bdnf*, which is also highly expressed in gut-innervating peripheral sensory peptidergic neurons emerging from the dorsal root ganglia [10,12,13]. The same increase in the expression levels of *BDNF* was observed in our analysis of human UC datasets. Interestingly, we also detected an elevation in the expression levels of markers for different neuronal subpopulations such as *NOS1* (inhibitory motor neurons) and *CALCB* (cholinergic sensory) in our analysis of the human UC datasets [10,12,13]. To the best of our knowledge, there has been no previous study reporting such a diverse and generalized increase in various subpopulations of neurons in the context of colitis.

Other than the increase in the neuronal populations in colitis, our analysis also showed an increase in the expression of various glial cell markers during colitis. Unfortunately, unlike enteric neurons, it has proved difficult to functionally categorize enteric glial cells into distinct subpopulations. However, previous studies have highlighted the role of different enteric glial cell populations in maintaining homeostasis in the gut [35,36]. Our analyses of human datasets revealed an increase in the expression levels of the glial cell-derived neurotrophic factor (*GDNF*) in UC patients. This is interesting given the discovery by Ibiza et al., which showed that ligands of the GDNF family derived from enteric glial cells were able to activate the RET receptor on type 3 innate lymphoid cells modulating the release of cytokines such as IL22, which are integral to epithelial protection in colitis [37]. The authors also showed that the expression of the GDNF family ligands by enteric glial is dependent on the sensing of the gut microenvironment in a manner dependent on toll-like receptor signaling intrinsic to enteric glial cells [37]. Interestingly, in our analysis of the human UC remission cohort, *GDNF* expression stayed high during remission and was negatively correlated with that of *S100A8*, implying that the glial-ILC3, GDNF- IL22 mechanism of tissue healing and barrier protection could play a significant role during remission in UC.

A general feature of inflammation is the loss of parenchymal tissues and infiltration and gain in stromal and mesenchymal cells, which combat and clear the threats and help in restoring the parenchymal cells (as depicted in Figure 4). The increase in the immune and neuroglial compartments in the inflamed tissues of IBD indicates a functional interaction of these two cellular compartments in regulating inflammatory processes. Indeed, we found a strong upregulation of several identifiers of neuroglia in the inflamed tissues of mouse models of colitis as well as human IBD samples. The upregulation correlated with the expression of the inflammation marker *S100A8*. In contrast to our findings, a previous study by Sanovic et al. found that experimental colitis induced using dinitrobenzene sulfonic acid (DNBS) resulted in damage and loss of the ENS, where neuronal numbers steadily fell over the course of 35 days post-DNBS treatment [16]. However, the authors were unable to correlate this with an increased TUNEL positivity or with other direct markers of cell death in the neuronal compartments but could show an increase in the number of infiltrating neutrophils and eosinophils in the myenteric plexi of DNBS-treated mice. Another study reported an increase in the levels of apoptotic enteric neurons in the inflamed ileal tissues of CD patients, but in stark contrast found a reduction in the number of apoptotic neurons in the colon of UC patients [4]. In the same study, the authors reported an increase in the number of apoptotic enteric glia in both CD as well as UC tissues. These data would indicate an elevation in the ongoing process of cell death in the enteric neuroglial compartment but do not describe the consequences of such cell death on the overall numbers of enteric neuroglia in IBD. These studies suggest that the enteric neuroglia are lost due to apoptotic cell death during the course of inflammation in IBD while our data suggest the opposite. Another hypothesis is that due to the ongoing inflammation, neuroglia suffer from damage and cell death, which trigger pathways of growth transdifferentiation, and neurogenesis in the ENS. Indeed, DSS-induced colitis was shown to trigger the transdifferentiation of SOX2+, PLP1+ glial cells to enteric neurons [14]. Furthermore, p75+ neural crest stem cells were shown to persist in the adult gut, and p75+, CD49b+ enteric glial cells were shown to exclusively undergo gliogenesis in the DSS- and non-steroidal anti-inflammatory drug-induced models of intestinal inflammation. By contrast, Belkind-Greson et al. showed that CD49b+ glial cells can undergo transdifferentiation to form HuC/D+ enteric neurons in response to DSS-induced colitis [14]. Therefore, it is likely that colitis in fact drives an expansion in the numbers of neuroglia in the intestine, similar to the observations presented here. Most studies used co-immunostaining of neuronal proteins together with BrdU as a common method to identify active neurogenesis in the ENS. However, this was put into question by the report of Joseph et al., who showed that BrdU incorporation is highly inconsistent in detecting adult ENS neurogenesis in multiple mouse models of inflammation at a wide range of ages and therefore previous data that relied only on this method should be interpreted with caution [18]. Our report uses a comparative transcriptomic approach and uses immunostaining to confirm the findings from transcriptomics to arrive at the conclusion of the elevated neuroglial repertoire in IBD. However, using this approach we are unable to identify whether or not there is active neurogenesis in the inflamed colonic tissues. This approach also does not address the functional impact that the increased neuroglial repertoire might have on the stool frequency, peristalsis, or the quality of life, and their correlation with neuroglial expansion, which needs to be evaluated in future studies. The upregulation of neuronal genes coding for structural proteins such as TUBB3 and the elevated innervation of the inflamed mucosa does indicate that inflammation induces growth-promoting changes in neurons. The fact that the elevated expression of the identifier genes returns to baseline levels during resolution and remission of colitis indicates a dynamic and rapid adaptive response and it is unlikely that the increase in cell density and innervation is permanent. Such dynamic behavior has been recently reported to occur in the ENS during homeostatic conditions [17]. It is therefore likely that inflammatory queues and DAMPs induce such neuroglial growth during inflammatory phases (Figure 4 summarizes, in brief, the cytokine and DAMP effects on ENS).

Taken together, it is now clear that intestinal inflammation triggers a local and dynamic increase in the neuroglial repertoire. This change is transient and subsides during remission and resolution. The exact nature of this dynamic response, the pathways involved, and the far-reaching functional consequence for neuroglial–immune and neuroglial–epithelial communication remain to be explored.

## Figures and Tables

**Figure 1 cells-10-02253-f001:**
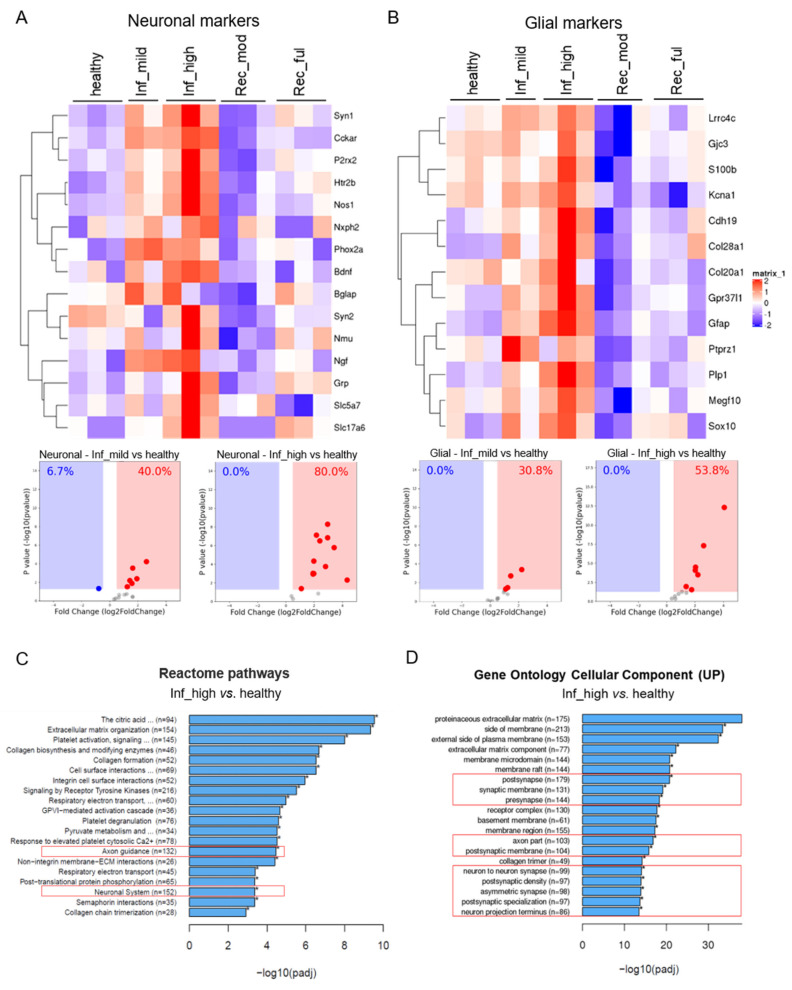
Elevated neuroglial signature in experimental colitis. Heatmap showing the normalized expression levels of (**A**) neuronal and (**B**) glial identifiers at various stages of DSS-induced colitis and its resolution representing the states: healthy, mildly inflamed (Inf_mild), highly inflamed (Inf_high), moderately recovered (Rec_mod), and fully recovered (Rec_ful). The volcano plots in A and B represent significant log_2_ fold changes of the identifiers in the Inf_mild vs. healthy and the Inf_high versus healthy groups. Each dot represents a gene, the red and blue solid rectangles represent zones of significance for up- and downregulated genes, and the numbers represent the percentage of identifiers found in the respective zone. (**C**) Reactome pathways and (**D**) gene ontology for cellular components regulated between Inf_high versus healthy groups. *n* = 3 for all groups except Inf_mild, where *n* = 2.

**Figure 2 cells-10-02253-f002:**
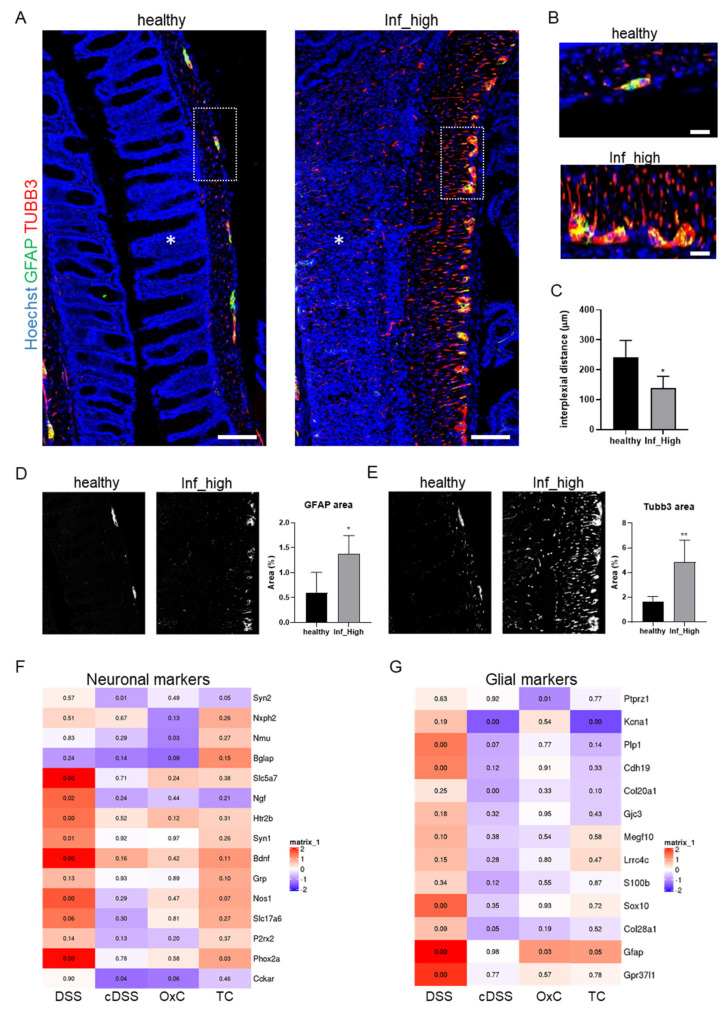
Innervation to the inflamed colonic mucosa is elevated. (**A**) Representative pictures of colonic tissue sections from healthy and highly inflamed (Inf_high) groups from the DSS-induced colitis experiment that were immunostained for GFAP (green) and TUBB3 (red) and counterstained with Hoechst 33,342 (blue). Asterisks denote mucosal regions; scale bars = 100 µm. (**B**) Insets from A showing zoomed-in views of the myenteric plexi. Scale bars = 50 µm. (**C**) Mean ± STDEV of interplexial distances in microns for the indicated groups. (**D**,**E**) Representative single-channel threshold images for the quantification of areas positive for (**D**) GFAP and (**E**) TUBB3 from the healthy and the DSS-induced colitis groups with high inflammation (Inf_high). For (**A**–**E**), healthy (*n* = 4) and Inf_high (n = 6). * *p* < 0.05, ** *p* < 0.01 in unpaired two-sided *t*-test. Heatmaps from RNA Seq data showing normalized fold changes (versus respective controls) in the expression levels of (**F**) neuronal and (**G**) glial identifiers in various mouse models of colitis including acute DSS-induced colitis (DSS), chronic DSS-induced colitis (cDSS), Oxazolone-induced colitis (OxC), and T cell adoptive transfer-induced colitis (TC) with n = 3 for each model and numbers in the boxes represent *p*-values. Note that samples for T cell transfer colitis were normalized against read counts from colons of Rag1^-/-^ mice (untreated control mice).

**Figure 3 cells-10-02253-f003:**
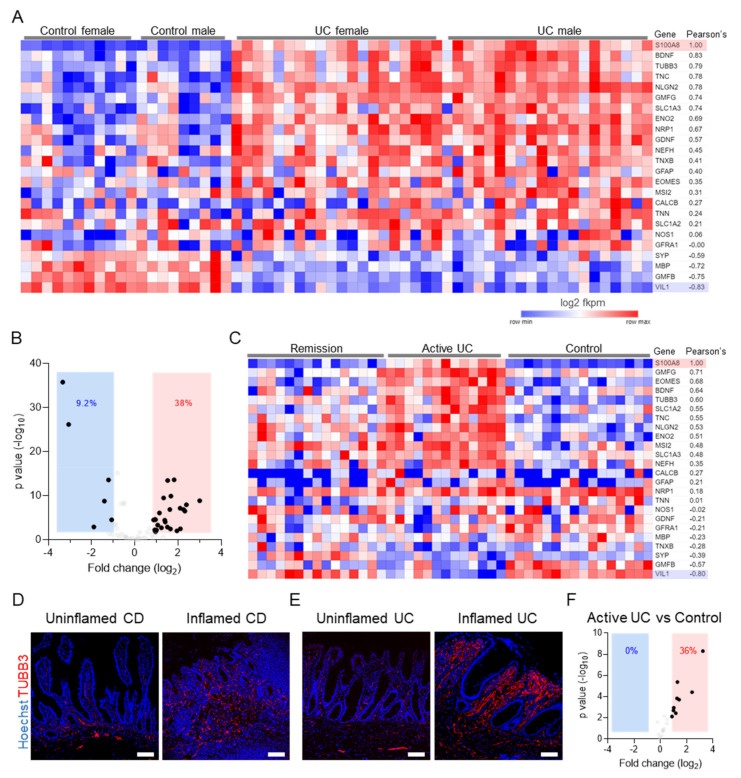
Inflamed tissues of IBD patients show increased neuroglial signature and innervation. (**A**) Heatmaps showing normalized log2 fkpm counts from the publicly available PROTECT ulcerative colitis (UC) dataset, showing the expression levels of the indicated neuroglial identifiers and their correlation with the inflammation marker S100A8 (red highlight). The expression of the epithelial marker VIL1 (blue highlight) was used as a reference for epithelial erosion. (**B**) Volcano plots for genes shown in A, representing significant fold changes in UC versus controls combined for both sexes. (**C**) Heatmaps showing normalized log2 fkpm counts of the neuroglial identifiers and their expression correlation with the inflammation marker S100A8 (red highlight) from the publicly available dataset of UC patients either in remission or with active disease and controls. The expression of the epithelial marker VIL1 (blue highlight) was used as a reference for epithelial erosion. (**D**,**E**) Sections of the uninflamed and inflamed intestine of (**D**) CD patients and (E) UC patients, immunostained with TUBB3 (red) and Hoechst 33,342 (blue). Scale bars = 100 µm. (**F**) Volcano plots for neuroglial identifiers representing significant fold changes in active UC versus controls from the independent patient cohort. In B and F, each dot represents one gene, the red and blue solid rectangles represent zones of significance of up- and downregulated genes, and the numbers represent the percentage of total identifiers found in either of these significance zones.

**Figure 4 cells-10-02253-f004:**
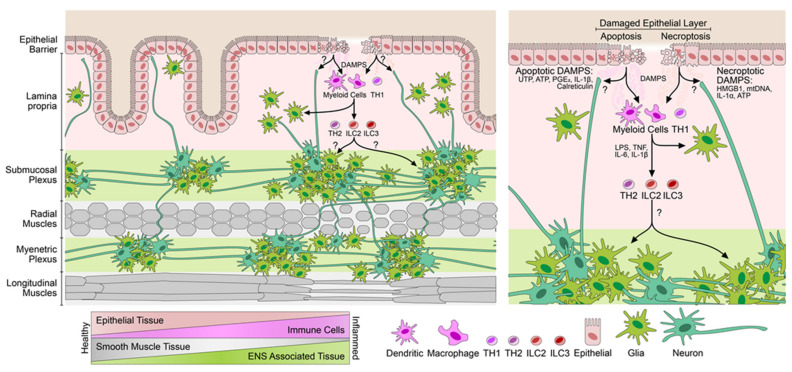
Consequences of inflammation in the gut on enteric nervous architecture. The schematic on the left depicts the elevated neuroglial innervation and repertoire in the inflamed tissues of the gut. The damage-associated molecular patterns (DAMPs) from dead and damaged cells trigger either direct (axon migration) or indirect (immune-mediated) neuronal innervation of the inflamed areas. Immune mediators and DAMPs promote the growth of the underlying neuroglial tissues in colitis, which subsides during remission. Cellular changes in the inflamed intestine are depicted below. Inflammation correlates with increased immune and enteric neuroglial repertoires, whereas the numbers of epithelia and muscles (in case of fistulation) are reduced. A zoom-in on the right depicts changes in the epithelial–immune–neuroglial homeostasis during inflammation with various mediators acting on different cell types.

## Data Availability

The data from the RNA sequencing experiments used here have been deposited to the public platform array express EMBL-EBI through the accession number E-MTAB-9850. All other datasets analyzed in this study along with their accession numbers, references, and sources for the data are included in the Section 2 of this article.

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
