# Peer review of "Dynamic, Transient, and Robust Increase in the Innervation of the Inflamed Mucosa in Inflammatory Bowel Diseases"

_cells, 2021, doi:10.3390/cells10092253_

Round 1

Reviewer 1 Report

The paper presents interesting results. Please provide more detailed discussion attempting to explain the source of differences with previous reports and to presume implications of your findings.

Minor concerns:

Please describe in methodology which experimental models of colitis were used (providing citation is not enough for reader)

Please specify the role of people who are acknowledged for the research. 

Please add number of Bioethical Consent for the study. 

Author Response

Reviewer 1:

The paper presents interesting results. Please provide a more detailed discussion attempting to explain the source of differences with previous reports and to presume implications of your findings.

Response: We are glad that the reviewer finds our work interesting. We would also like to thank the reviewer for the time and for reviewing our work. As requested, we have now included a paragraph comparing previous reports and the broader implications of our findings (revised manuscript, pages 12 and 13, lines 304 to 328)

Minor concerns:

Please describe in methodology which experimental models of colitis were used (providing citation is not enough for the reader)

Response: We have now provided the necessary details on the experimental models of colitis as requested by the reviewer (revised manuscript page 3, Lines 86- 97)

Please specify the role of people who are acknowledged for the research.

Response: Details on the roles of the people that appear in the acknowledgments have been provided (revised manuscript page 15, Lines 395- 96).

Please add a number of Bioethical consents for the study. 

Response: The approval numbers for the studies presented in our work have now been added (revised manuscript page 15, lines 386 to 388 and page 3, line 83)

Reviewer 2 Report

Acera et al represent the changes of the ENS in IBD in their study. The study contains multiple new aspects in th field of IBD. There are some minor comments:

Changes in the enteric neuroglia have been visualized in the DSS model. Are these data for the other mouse models eg oxazolone- 196
induced colitis, and T-cell adoptive transfer coliti
avaible as well?

Is the increase of the nueroglial identifiers in humans associated with more stomach discomfort or a higher stool frequence? are any data on QoL of these patients avaiable?

Is the ENS downregulated again after an IBD flare in mice/human?

Author Response

Please see the attachment for the responses to reviewers comments.
